# Bitcoin Network Mechanics: Forecasting the BTC Closing Price Using Vector Auto-Regression Models Based on Endogenous and Exogenous Feature Variables

**Ahmed Ibrahim [1], Rasha Kashef [2,*], Menglu Li [2], Esteban Valencia [3] and Eric Huang [4]**

[1] Computer Science Department, Wilfried Laurier University, Waterloo, ON N2L 3C5, Canada; ahibrahim@wlu.ca

[2] Electrical, Computer, and Biomedical Engineering Department, Ryerson University, Toronto, ON M5B 2K3, Canada; menglu.li@ryerson.ca

[3] IVEY Business School, Management Science Department, London, ON N6G 0N1, Canada; evalencia.msc2017@ivey.ca

[4] Advanced Analytics and Research Lab, Toronto, ON M5J 2P1, Canada; ehuang.msc2017@ivey.ca

* Correspondence: rkashef@ryerson.ca

**Abstract:** The Bitcoin (BTC) market presents itself as a new unique medium currency, and it is often hailed as the "currency of the future". Simulating the BTC market in the price discovery process presents a unique set of market mechanics. The supply of BTC is determined by the number of miners and available BTC and by scripting algorithms for blockchain hashing, while both speculators and investors determine demand. One major question then is to understand how BTC is valued and how different factors influence it. In this paper, the BTC market mechanics are broken down using vector autoregression (VAR) and Bayesian vector autoregression (BVAR) prediction models. The models proved to be very useful in simulating past BTC prices using a feature set of exogenous variables. The VAR model allows the analysis of individual factors of influence. This analysis contributes to an in-depth understanding of what drives BTC, and it can be useful to numerous stakeholders. This paper's primary motivation is to capitalize on market movement and identify the significant price drivers, including stakeholders impacted, effects of time, as well as supply, demand, and other characteristics. The two VAR and BVAR models are compared with some state-of-the-art forecasting models over two time periods. Experimental results show that the vector-autoregression-based models achieved better performance compared to the traditional autoregression models and the Bayesian regression models.

**Keywords:** Bitcoin; blockchain; autoregression; time-series analysis; simulation; predictive modes; endogenous; exogenous variables

## 1. Introduction

Bitcoin (BTC) is a digital currency alternative to real currency and is the most popular among the cryptocurrencies. The BTC was created by a cryptologist known as "Satoshi Nakamoto", whose real identity is still unknown (Nakamoto 2008). As blockchain currencies are not as liquid as other forms of currency, understanding the behavior of this market draws insights as to how one could capitalize on this asset over time. Especially as society becomes more digitally inclined, the viability of a blockchain currency such as BTC to become a common currency seems like a possible reality. There are both winners and losers in the context of each capital market transaction. There are several drivers impacting

the Bitcoin market, such as the total number of Bitcoin available, the difficulty of Bitcoin mining, and average blockchain size. Therefore, determining the essential endogenous and exogenous drivers in BTC markets is a critical task. Each of these endogenous and exogenous variables can be treated as a time series, and therefore suitable multivariate time series forecasting models are needed.

Vector autoregression (VAR) is one of the most widely-used stochastic process models to analyze interdependencies of multivariate time series, and it has proven to be a useful model to describe the behavior of economic and financial time series, and to forecasting (Campbell et al. 1996). The VAR model is an extension of the univariate autoregression model to multivariate time series data. In the VAR structure, each variable is a linear function of past lags of itself and the past lags of the other variables. However, the limited length of standard economic datasets may produce over-parameterization problems (Koop and Korobilis 2009) thus, the Bayesian vector autoregression (BVAR) model was introduced in Litterman (1980) to solve this problem. The BVAR model uses Bayesian methods to estimate a vector autoregression. In comparison with the standard VAR models, the BVAR model treats input parameters as random variables, and prior probabilities are then assigned. A feature-selection of the cryptocurrency drivers is strongly needed to enhance the performance of a multivariate time-series (e.g., BTC) prediction model. In this paper, we applied direct forecasting using VAR and BVAR models to simulate the BTC market to understand the behavior of market participants as well as their most and least favorable market conditions according to the closing price of BTC based on an optimal set of exogenous variables. The simulated BTC market includes forecasting the endogenous variables, such as the equilibrium closing price of the market for BTC as denominated by the US dollar (MKPRU), the number of unique MyWallet users (MWNUS), and the total BTC available in the market to date (TOTBC). Experimental analysis over 7-year and 10-year timeframes shows the efficiency of the VAR and BVAR models in predicting the set of endogenous variables compared to traditional autoregression and Bayesian regression models using the optimal selected set of exogenous variables. The rest of this paper is organized as follows: Section 2 introduces the background of Bitcoin; Section 3 focuses on the related work; Section 4 describes the prediction models for Bitcoin closing price; Section 5 presents and discusses the results of the prediction models; and Section 6 outlines the conclusions and future works.

## 2. Background on Bitcoin

Bitcoin is a unique digital currency with the potential to change the nature of the transactions that people conduct in digital space. Bitcoin enables consumers for the first time to make electronic transactions from person to person without the need for an intermediary between them, like cash (Brito 2014). Transactions conducted in the digital space with BTC allow individuals to push payments directly to the merchants without having to share personally identifiable information, which could be intercepted by cybercriminals for fraud. One of the greatest concerns for BTC as a commonly accepted currency is the security, as there is no intermediary to ensure the coverage on stolen BTC, should theft occur (Brito 2014). As the value of the asset appreciated 63% YTD in 2016, and 87% YTD in 2020, identifying historical patterns of behavior could help in understanding how the BTC security (and the security of similar cryptocurrencies) is likely to behave from inception.

### 2.1. Bitcoin Ledger

Each block in the Bitcoin blockchain contains a summary of all transactions in the block using a Merkle tree (aka binary hash tree) such that each transaction is first put into a pool of pending transactions. Then, they are put into the transaction chain (blockchain) (Antonopoulos 2014). Each block is linked in a chain by a reference to a previous header hash in which the addition of a transaction into the chain is through a "mathematical lottery" (United States Securities and Exchange Commission 2017). The miner solves the math problem (cryptographic hashing) and puts the transaction into the chain. The math helps everyone with a wallet know the order of transactions as well as all past transactions.

*2.2. Bitcoin Development Process*

As other cryptocurrencies aim to perform the same computer distributed task, there are risks that any new digital currency faces from inception until maturity. There are three primary characteristics that a digital currency must satisfy to be deemed a sound form of currency. The following are the key success factors (Barski and Wilmer 2015):

- The network effect;
- Cryptocurrency volatility;
- Cryptocurrency-pegging technology.

2.2.1. The Network Effect

The simple concept of money is that people will be willing to use the currency (medium of exchange) so long as someone else is willing to accept it as a form of payment. Without an appropriate network for the payment mechanism, it is unlikely that people would desire to use the specific cryptocurrency if it turns out to be illiquid.

2.2.2. Cryptocurrency Volatility

For any cryptocurrency that is getting newly established as a payment method, the "fair" established value must be stable for consumers to be comfortable purchasing with the digital currency. As BTC is a newly available asset, the price discovery mechanism requires that the group of buyers and sellers exchanging the currency come to an agreed-upon value for the underlying asset (Pagnottoni and Dimpfl 2019). As the value of a BTCUSD in November 2016 was roughly $740, the currency was far from stable at the time. Seeing prices as high as $1200 in 2013, $15,000 in 2020, and as low as $355 in 2016 for BTCUSD, a true concern for consumers is to make a purchase with an asset that has varied so much in value. However, there are many cases of money being just as volatile. One famous example being the Zimbabwe hyperinflation, where the currency experienced 80 billion percent inflation in a single month.

2.2.3. Cryptocurrency-Pegging Technology

As the supply for the total BTC is limited to 21,000,000, more users have begun to use the BTC, which has modestly reduced volatility. The advantage of BTC over other cryptocurrencies is that it has been established and generated credibility for a sufficient network of users to adopt the use of the coin. Primarily, this has helped BTC outpace other digital currencies to normalize volatility. For any potential new e-coin that could enter the cryptocurrency market, it would make sense that the coin merges its stability according to a more stable cryptocurrency such as BTC.

*2.3. Market Participants*

The following are the market participants worthy of further analysis accompanied by a brief description of their role in the market:

- Miners—The market participants who are proactively adding transaction records to Bitcoin's public ledger of past transactions or blockchain and fueling the supply of BTC.
- Individual investors—Investors for the digital assets to purchase goods or services with the digital currency.
- Payment mechanism—Conduct business internationally as international payments are now available via BTC.
- Retail investors—Funds that are likely to pick up the currency as a portion of their portfolio to hedge, like gaining exposure to traditional currency markets.

*2.4. Stakeholders*

As digital currency changes the evaluated value of money and other financial assets, several stakeholder requirements and motives should be considered. The following are the stakeholders (formal and informal) affected by the adoption of cryptocurrencies: savers/bullish investors, government, other cryptographers, BTC exchanges/brokers, illegal black markets, BTC miners, and members of the public. As stakeholders desire stability and strength with any medium of transaction, some stakeholders are opposed to the widespread adoption of BTC. Specifically, the government and other cryptographers may have an issue with the widespread adoption of the BTC as decentralized digital money where no government or single entity can control the price or value.

## 3. Related Work

In modeling and simulation of the economics of mining in the Bitcoin market (Cocco and Marchesi 2016), authors have discussed how a miner is impacted by BTC prices (Cocco and Marchesi 2016). The goal of this artificial market model is to model the economy of the mining process from the inception of the Graphics Processing Units (GPU) generation. The important findings for this computational experiment encompass the ability to reproduce the unit root property, the fat tail phenomenon, and the volatility clustering of the BTC prices (Cocco and Marchesi 2016). Research on Bitcoin price forecasting are mainly based on two approaches: machine learning and time series methods.

*3.1. Machine Learning Prediction Methods*

Felizardo et al. (2019) presented a comparative study of price prediction performance among several machine learning models: long short-term memory (LSTM), WaveNet, support vector machine (SVM), and random forest (RF). The results indicated that for time-series data, the LSTM model tends to perform better than other machine learning models. The research of Tandon et al. (2019) gave a similar conclusion. They applied three different machine-learning methods to forecast the Bitcoin price, and compared their prediction ability. As a result, the RNN (recurrent neural network) with LSTM gave a lower mean absolute error than the random forest and linear regression models. Many research focuses on improving the LSTM model to increase forecasting accuracy. Wu et al. (2018) proposed an LSTM called LSTM with AR(2) model to forecast Bitcoin's daily price. The conventional LSTM model only considers the previous price of to predict the current price; instead, the LSTM with AR(2) takes the previous two days' prices into account. The experimental results demonstrated that the proposed model with AR(2) achieved a better forecasting accuracy with a lower mean squared error. Hashish et al. (2019) proposed the addition of hidden Markov models (HMMs) to the conventional LSTM. The HMM was used to describe the historical movements of Bitcoin. The proposed hybrid of HMM and LSTM outperformed the traditional forecasting of LSTM by decreasing the mean squared error from 49.089 to 33.888. The main drawback of the machine-learning models is that these models need high computational capacity, and so the execution time of the forecasting process is very time consuming. Thus, in this paper we focus on time-series prediction models. Support vector machine, latent source, and multilayer perceptron models work better for classification problems. The LSTM model performs well in solving long-term dependency problems, which means it is suitable for price prediction. However, the LSTM model needs a long computation time and has a large memory requirement.

*3.2. Time-Series Prediction Methods*

Bakar and Rosbi (2017) proposed the autoregressive integrated moving average model (ARIMA) to forecast the exchange rate between Bitcoin and the US dollar. In this method, the upcoming price depends upon autoregression, integration, and moving average, respectively. They believed the ARIMA model could be a reliable model to forecast the volatile characteristic of Bitcoin. Both Roy et al. (2018)

and Anupriya and Garg (2018) applied the ARIMA model to predict Bitcoin's price. The experimental result demonstrated the strong forecasting ability of the ARIMA. The mean error between the actual prices and the predicted prices was less than 6% for most values. Roy et al. (2018) also compared the performance of the ARIMA model with the autoregressive model (AR) and moving average model (MA), and the ARIMA model resulted in better accuracy than the other two models. However, the ARIMA model's shortcoming is that this it can give a more accurate prediction for short-term data, based on the research result of Ariyo et al. (2014). Rane and Dhage (2019) introduced nine approaches for Bitcoin price prediction and discussed each methodology in their research. The ARIMA model targets to forecast uncertainty time-series data within a short-term period, but class imbalance can bias it. Linear regression is unsuitable to predict Bitcoin price as the time series data.

The strength of the vector autoregression (VAR) model and the Bayesian vector autoregression (BVAR) model to estimate currency and exchange rate fluctuations have been demonstrated in recent research. VAR has been used widely by financial theorists and economists in predicting time series economic variables in systems that involve supply and demand (Ito and Sato 2006; Wang et al. 2017; Carriero et al. 2009; Alquist et al. 2013; Sims 1993). We found several papers that use VARs to estimate currency and exchange rate fluctuations, notably Koray and Lastrapes, who use a VAR model to estimate the exchange rate on a series of macroeconomic variables (Koray and Lastrapes 1989). Additionally, Ito and Sato performed VAR research on the exchange rate of post-crisis Asia (Ito and Sato 2006). Wang et al. (2017) established a VAR model to analyze the impact of exchange rate volatility on economic growth. Furthermore, there is some research on forecasting using the Bayesian vector autoregression (BVAR) method. For example, Carriero, Kapetanio, and Marcellino demonstrated that the BVAR model produced better forecasting for exchange rates (Wang et al. 2017). In the econometric/finance community, (Catania et al. 2019) and (Bohte and Rossini 2019) have studied the forecasting performance of cryptocurrencies by vector autoregression with and without time-varying volatility. (Bianchi) has investigated the possible relationship between returns on cryptocurrencies and traditional asset classes. Bianchi et al. (2020) discussed the relationship between the returns on stable-coins and major cryptocurrency pairs within the context of a large Bayesian vector autoregression model. The BVAR model extends the classical VAR model by using Bayesian methods to estimate a vector autoregression. The BVAR model treats input parameters as random variables, and prior probabilities are then assigned. Current related work to both VAR and BVAR models in forecasting BTC prices does not focus on selecting the set of endogenous and exogenous variables and drivers that control the BTC market, which is the primary focus of this paper.

## 4. BTC Closing Price Prediction Models

Both VAR and BVAR models are used in this paper to forecast the Bitcoin price and simulate the BTC market to understand market participants' behavior as well as the market conditions according to the closing price of BTC.

### 4.1. Endogenous and Exogenous Variables

An autoregressive model is typically used to develop predictions and understand the trend of a time series. However, in financial and economic data, several factors are affecting the time series, such as supply, demand, and regulation. The complex nature of any financial market warrants a more sophisticated model. The performance of the VAR and BVAR forecasting models depends on the optimal selection of the set of endogenous variables of interest. Several variables were tested as proxies to represent the price, demand, and supply of the BTC market, respectively, after trying out numerous iterations of VARs and BVARs and using sensitivity analysis with different variables, lags, and time frames. The final set of endogenous variables is defined in Equation (1). Let $Y_t$ be a vector of the endogenous variable of interest such that:

$$Y_t = \{\text{MKPRU, MWNUS, TOTBC}\} \tag{1}$$

$$t_1 = [04\text{-}01\text{-}2009, 22\text{-}11\text{-}2016] \tag{2}$$

$$t_2 = [01\text{-}01\text{-}2011, 01\text{-}08\text{-}2020] \tag{3}$$

where MKPRU represents the equilibrium closing price of the market for BTC as denominated by the US dollar (Figures 1 and 2). MWNUS is the number of unique MyWallet users, and TOTBC is the total BTC available in the market to date, as there is a limited amount of BTC available at 21,000,000. Our time frames are across two intervals, the first one is [04-01-2009, 22-11-2016] (Figure 1), and the second period is [01-01-2011, 01-08-2020] (Figure 2). The decision-making process uses reasonable metrics deemed viable drivers of the endogenous variables, where the following were selected as the exogenous variables: Average Block Size in, MB (AVBLS), Bitcoin Difficulty (DIFF), Number of Transactions per Block (NTRBL), Miner's Revenue (MIREV), Change in the Number of unique addresses (NADDU), Total Output Volume (TRVOU), and Hash Rate (HRATE). A majority of the factors selected were those that had been a result of the BTC network's transaction behavior and how the fundamental mechanics influenced the closing price. The variables AVBLS, DIFF, TRVOU, and HRATE were taken as the variables which dictated the difficulty of accessing and supplying BTC to the market. NTRBL considers the growing number of transactions occurring per block of BTC as a measure of transaction volume per available block of BTC. The NADDU variable considers the changing number of unique addresses performing BTC transactions to understand behavior trends over time. TRVOU measures the exchange trade volume of USD within the BTC market, which serves as a guideline as to how the market reacts to changes in value when buying or selling BTC. Finally, $X_t$, as the list of exogenous variables, is defined in Equation (4) as:

$$Xt = \{\text{MIREV, NTRBL, AVBLS, DIFF, NADDU, TRVOU, HRATE}\} \tag{4}$$

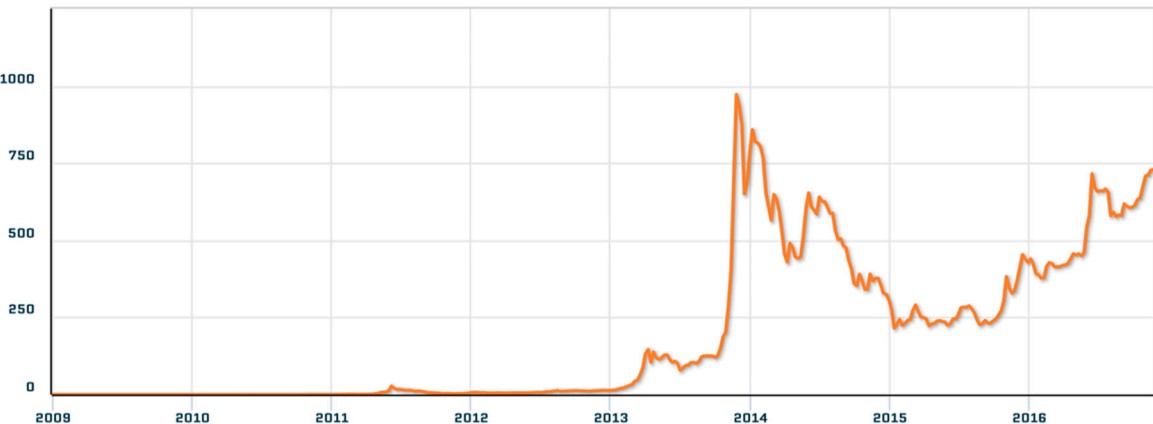

**Figure 1.** Bitcoin closing price in USD (MKPRU), [04-01-2009, 22-11-2016].

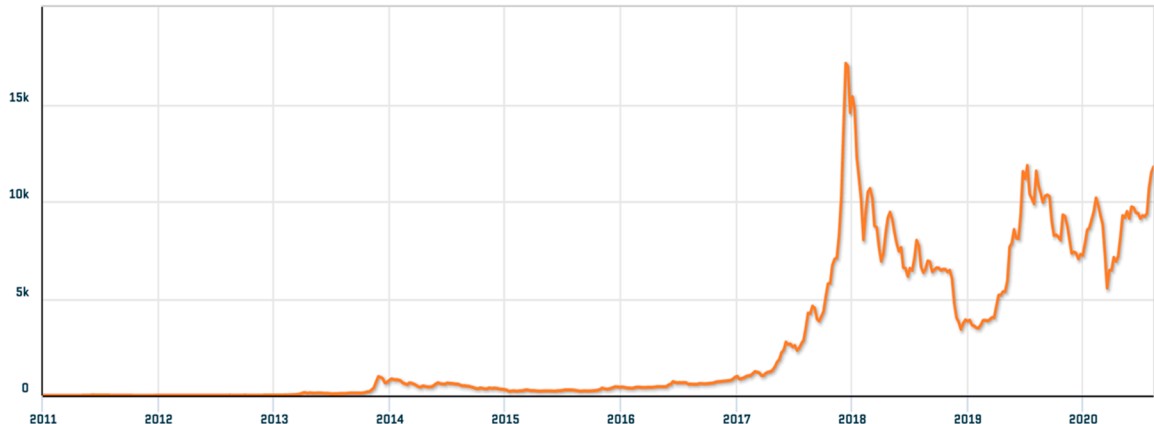

**Figure 2.** Bitcoin closing price in USD (MKPRU), [01-01-2011, 01-08-2020].

*4.2. Vector Autoregression (VAR) Model*

A Vector autoregression (VAR) (Sims 1993), (Kuschnig et al. 2020), and (Kuschnig and Vashold 2019) model was developed to understand the relationship between the system of variables that are of interest (Equations (1) and (4)). Thus, the VAR of interest is as follows:

$$Y_t = \beta_0 + \beta_1 Y_{t-1} + \beta_2 Y_{t-2} + \beta_3 Y_{t-3} + \cdots \beta_n Y_{t-n} + \beta_{n+1} X_t + \epsilon_t \tag{5}$$

where the betas ($\beta'_t s$) are vectors of constants and coefficients representative of the relationship between the variables, where $n$ is the number of lags used in the VAR model. The purpose of selecting this model is to use the model coefficients to simulate a certain period of BTC endogenous variable (Equation (1)) given the exogenous variables (Equation (4)). Furthermore, one could ideally forecast out the BTC price behavior over time, such that there are verified and validated forecasts of the exogenous variables.

4.2.1. Model Assumptions

A few assumptions were made in this VAR model in an effort to use real market data to forecast just over six months. First, the model assumes that the relationship between the variables is static. A variety of timelines were tested accordingly in order to understand differences in behavior. The following are the timeframes selected for analysis:

Experiment A: Full timeframe: [04-01-2009, 22-11-2016], Post-boom timeframe: [10-12-2013, 22-11-2016], the Year of 2016 timeframe: [01-01-2016, 22-11-2016]. Experiment B: Full timeframe: [01-01-2011, 01-08-2020], Post-boom timeframe: [01-01-2017, 01-08-2020], the Year of 2020 timeframe: [01-01-2020, 01-08-2020]. For both Experiments A and B, the second assumption made in the model was the segregation of endogenous and exogenous variables. The decision-making process yielded a qualitative and intuitive measure for the variables.

4.2.2. Model Validation and Verifications

The process of validating the model was among the most difficult tasks throughout the entire process. Ultimately, the selected set of endogenous variable contained the BTC exchange rate, a variable for supply, and a variable for demand. Collectively, these variables help represent the market mechanics of Bitcoin. Based on the selected endogenous and exogenous variables, the following parameters were used:

- lag.max = 366—to accommodate a full year of seasonal behavior and trends;
- type = 'both'—to evaluate the deterministic regressors.

The resulting selection of a timeframe was selected according to Akaike Information Criterion (AIC), Schwarz Criterion (SC), Hannan Quinn (HQ), and Forecast Prediction Error (FPE). This screening

process served as a deterministic selection of the timeframe for the forecasting by encompassing summary statistics such as *p*-value and $R^2$ to verify the accuracy of the relationship that was being estimated. Additionally, other combinations of variables were attempted with exceptionally poor results. Most of the other variables that were included as an aggregate to those used in the model projected dramatic market crashes with negative asset value.

### 4.3. Bayesian Vector Autoregression (BVAR) Model

The classical VAR model may have over-parameterization problems because of the large number of parameters and limited availability of time-series datasets (Sims 1980); alternatively, the Bayesian vector autoregression model can be used. The BVAR model applies Bayesian methods to estimate a VAR and treats the VAR model parameters as random variables. It also assigns and updates the prior probabilities of both observed and unobserved parameters based on available data (Miranda-Agrippino and Ricco 2018). The BVAR model in this paper uses the same variables of interest in the VAR model as described in Section 4.2. Let $Y_t$ be a list of variables used in this BVAR model, such that:

$$Y_t = \{\text{MKPRU, MWNUS, TOTBC}\} \tag{6}$$

As in the VAR model, the BVAR model also assumes the chosen variables have static relationships and uses several different timelines to observe forecasting outputs. The BVAR model uses the same timeframes (Experiment A and Experiment B) used in the VAR model in order to compare their forecasting abilities.

Prior Specification

In the BVAR model, the informative prior probability distribution of the VAR coefficients ($\beta_t's$ in Equation (5)) can be assigned before observing the sample data. The Minnesota prior was introduced and developed by Robert Litterman and other researchers at the University of Minnesota (Litterman 1980), and was chosen in our BVAR model. This prior is based on the behavior of most macroeconomic variables, which is approximately a multivariate random walk model with drift. The parameters of the Minnesota prior are set as follows:

- Parameter λ with max = 5 and min = 0.0001, to control the tightness of the prior;
- Parameter α with max = 3 and min = 1, to manage variance decay with increasing lag order;
- var = 10,000,000, to set the prior variance on the model's constant.

## 5. Experimental Analysis

Real datasets of the Bitcoin market in three different timeframes were used in this paper across two different time periods, Experiment A, $t_1$ = [04-01-2009, 22-11-2016], and Experiment B, [01-01-2011, 01-08-2020]. For Experiment B, the data were normalized using the logarithm of each return variable. Both the VAR and BVAR models were applied and tested on these datasets to forecast the Bitcoin market price. The forecasting results were analyzed to evaluate the performance of our models.

### 5.1. Experimental Dataset

The primary source of data and information was the Quandl Dataset sourced from Blockchain.com (Quandl 2020). The source contains up to 32 datasets, including the BTC market price. Each dataset contains a time series for a variable. The secondary dataset was the average OHLC (open-high-low-close) candlestick values across multiple exchanges scraped from Rbitcoincharts.com (Bitcoin Charts 2020). Additionally, any of the transforms accepted were denoted upfront before the variable. In the circumstance of the BTC simulation, the Quandl transform applied was "diff", which implied the change over time depending on the frequency (i.e., daily frequency data would be sampled as daily frequency change of that variable). The OHLC candlestick chart data (Figure 3) were sourced directly

from Rbitcoincharts.com, consolidating the average OHLC candle according to a number of varying exchanges which trade BTC and similarly pegged altcoins. One of the major difficulties encountered upon sourcing the data was to get a consistent market price from BCHAIN, which would match the OHLC charts sourced. The difference appeared to be according to when the different data sources selected their end-of-day settlement. Rbitcoincharts.com was selected, as the close price difference was roughly around ($1–$2).

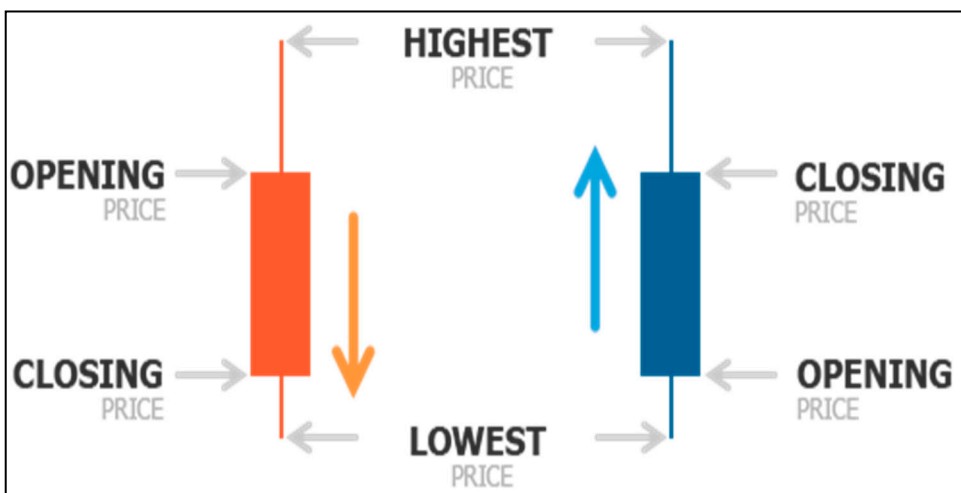

**Figure 3.** OHLC (open-high-low-close) candlestick.

*5.2. Forecasting Results*

Both VAR and BVAR models were tested with three timeframes in two different experiments. Experiment A: For the 2016 timeframe, values of variables described in Sections 4.2.1 and 4.2.2 between 01-01-2016 and 30-09-2016 were imported as input to the two models. For the Post-boom timeframe, data from 10-12-2013 to 30-09-2016 were imported as input. Both models forecasted the Bitcoin price in USD for the period 01-10-2016 to 30-10-2016 and compared the forecasting results with the actual Bitcoin price. For the Full timeframe, the time period selected to forecast was the last 199 days [05/08/2016–11/22/2016] to evaluate the effectiveness of these two models. Experiment B: For the 2020 timeframe, input and output variables between 01-01-2020 and 01-08-2020 were used for both the VAR and BVAR models. For the Post-boom timeframe, data from 01-01-2017 to 01-08-2020 were used. For the Full timeframe, the time period selected for forecasting was the last six months [01-02-2020, 01-08-2020] to evaluate the effectiveness of these two models.

5.2.1. Results of the VAR Model: Experiment A

The model selects the most suitable coefficients, where the outcome minimizes FPE. Figures 4–6, respectively show the evaluation of the Full, Post-boom, and the Year of 2016 timeframes forecasting in comparison to the BTCUSD OHLC candle from Rbitcoincharts.com, where "fcst" is the forecasted closing price, "lower" is the lower bound (95% CI), and "upper" is the upper bound (95% CI). The endogenous variables were simulated from the estimated VAR, as shown in Figures 7–9 for three different timeframes. The simulated exogenous variables were the real datasets taken from Quandl for the aforementioned timeframe. Ultimately, by evaluating the results of different timeframes, the full timeframe using the VAR model showed the best forecasting performance. The Full timeframe represents the most data available and incorporates the relationships over different timeframes. Although the significance of the relationship between these variables may change over time, the 7-year timeframe surely aided in modeling the market behavior.

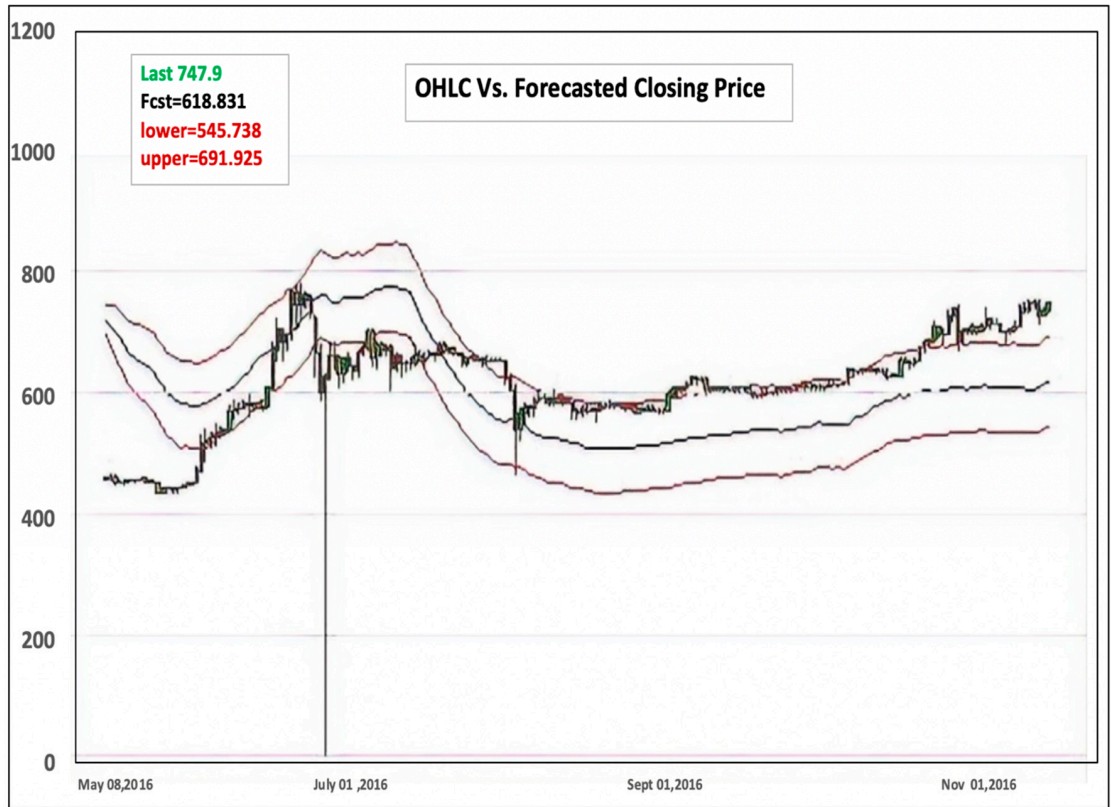

**Figure 4.** Forecasting Bitcoin closing price using Full timeframe. Data Vs. BTC OHLC.

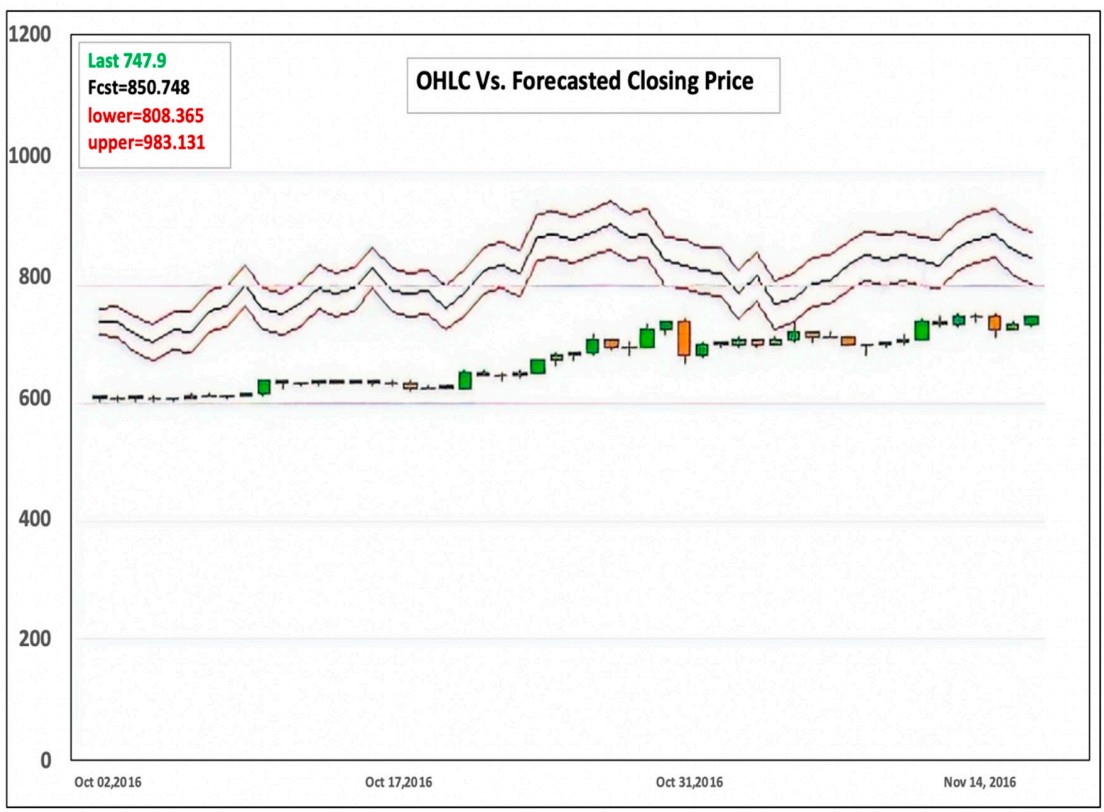

**Figure 5.** Forecasting Bitcoin closing price using Post-boom timeframe. Data Vs. BTC OHLC.

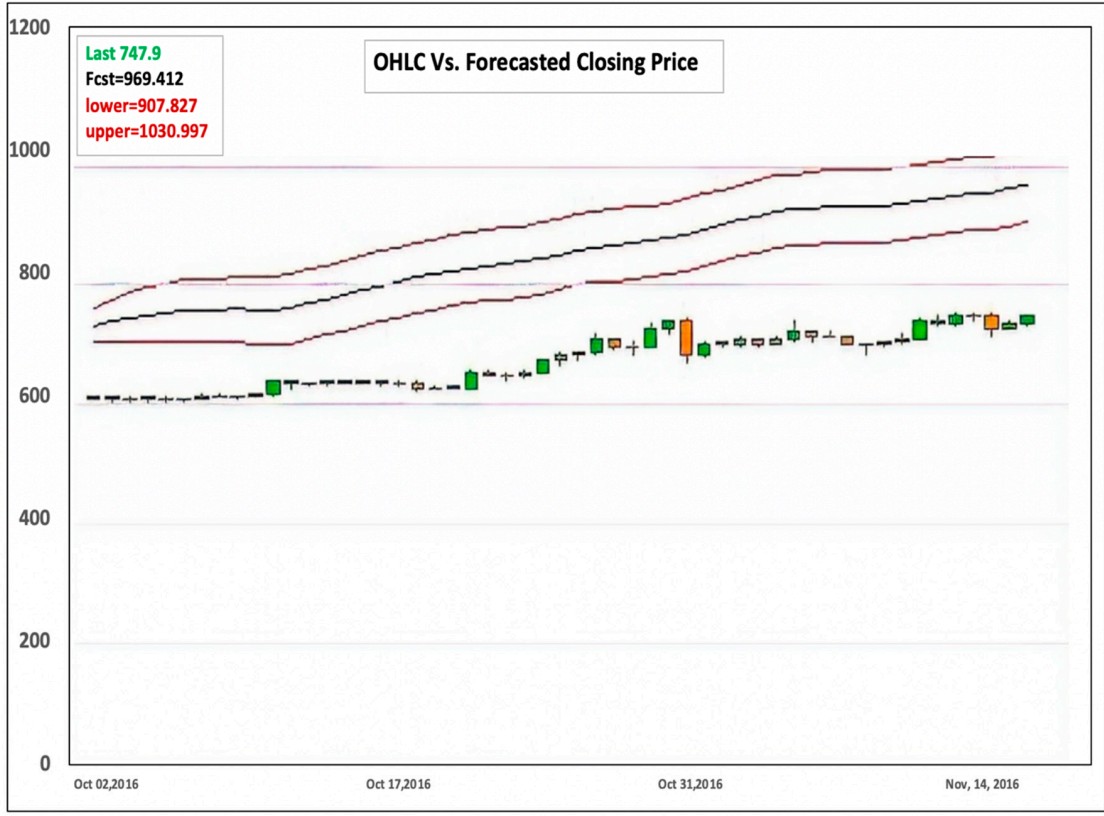

**Figure 6.** Forecasting Bitcoin closing price using Year of 2016 timeframe. Data vs. BTC OHLC.

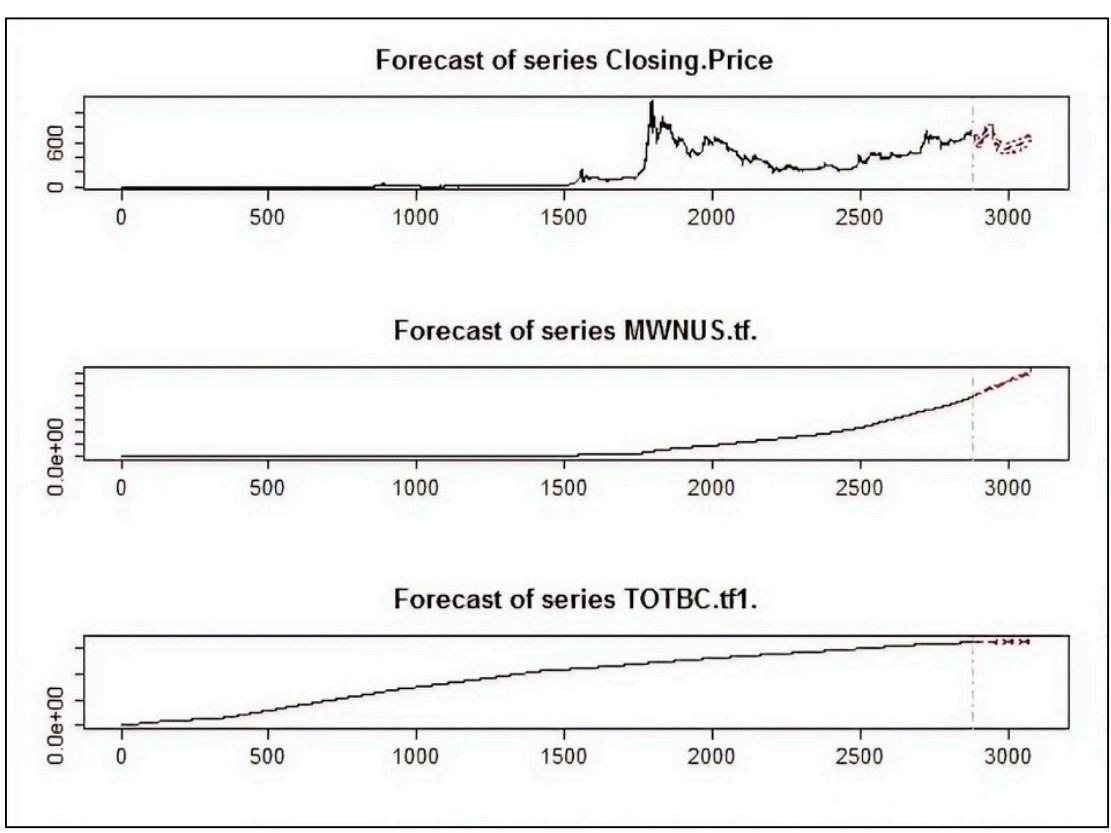

**Figure 7.** Forecasting the endogenous variables using Full timeframe data (VAR).

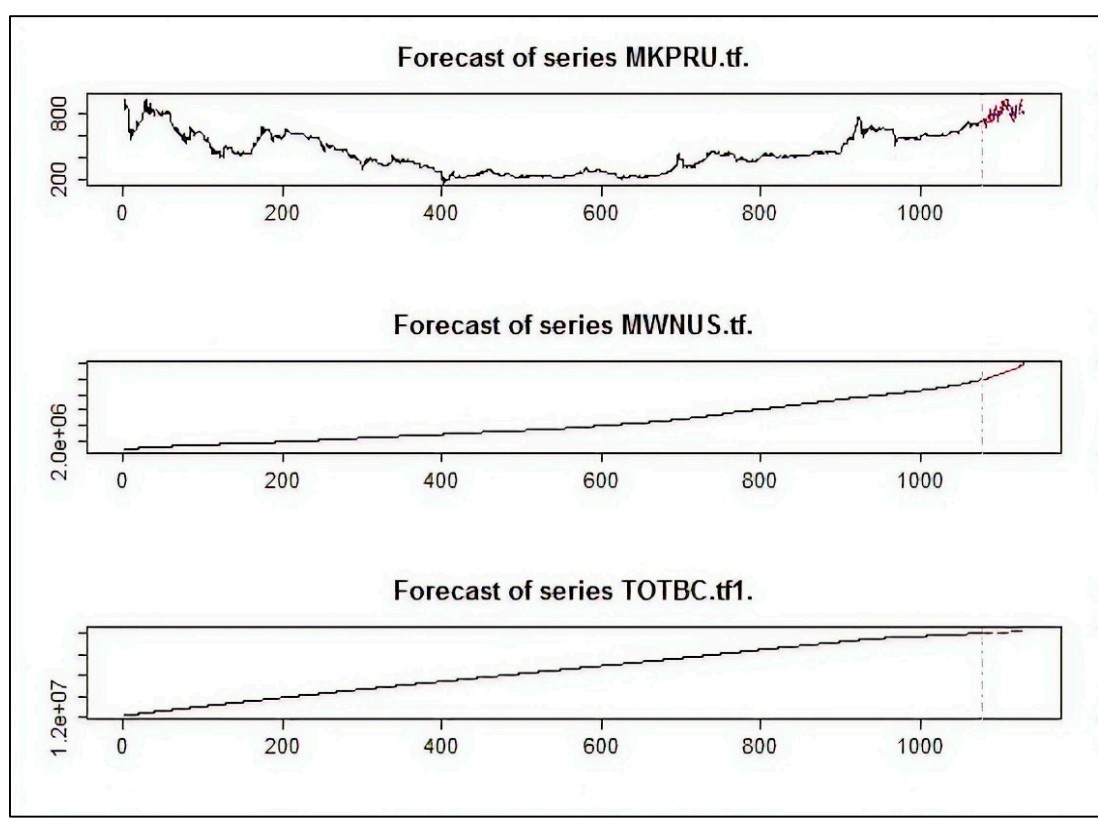

**Figure 8.** Forecasting the endogenous variables using Post-boom timeframe data (VAR).

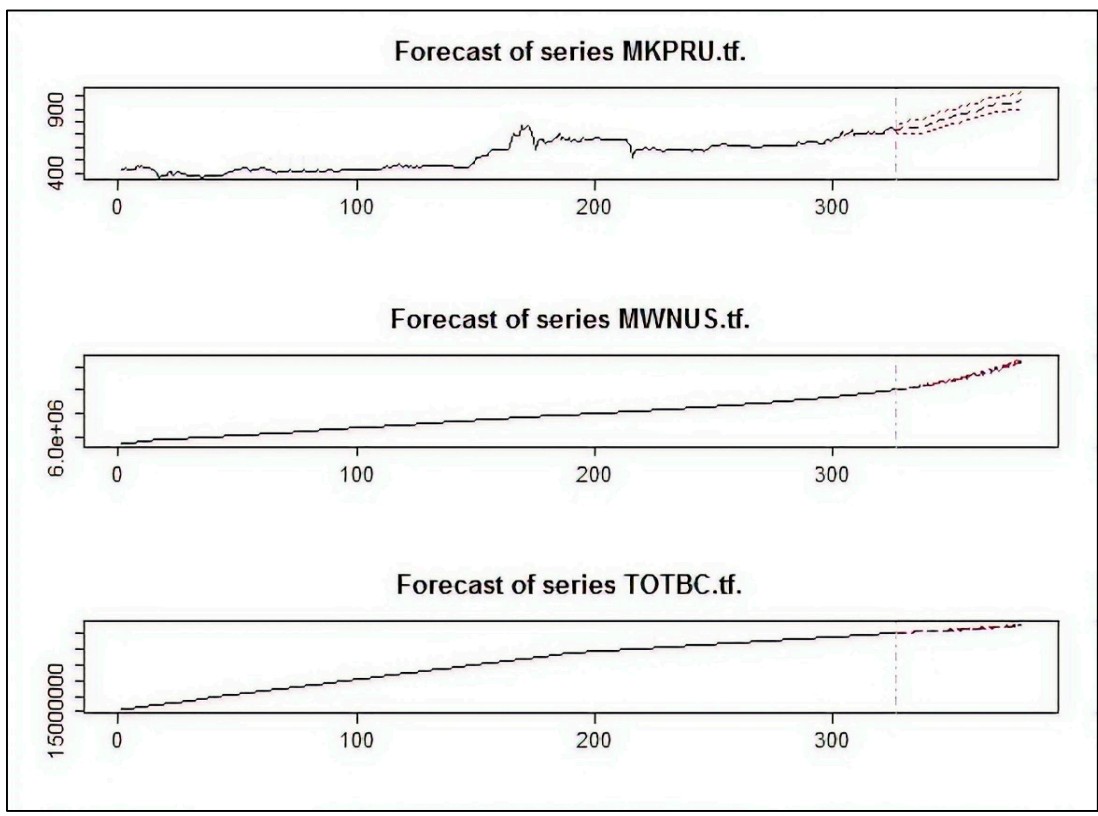

**Figure 9.** Forecasting the endogenous variables using Year of 2016 timeframe data (VAR).

### 5.2.2. Results of the VAR Model: Experiment B

In this experiment, we evaluated the performance of the VAR model using the period [January 2011–August 2020] Full timeframe data, Post-boom timeframe data [January 2017–August 2020], and the Year of 2020 timeframe data [January 2020–August 2020]. We can observe that the VAR model could effectively predict the prices of the BTC using the three timeframes for the variables MKPRU, MWNUS, and TOTBC, as shown in Figures 10–13, with the best performance obtained for the Full timeframe period.

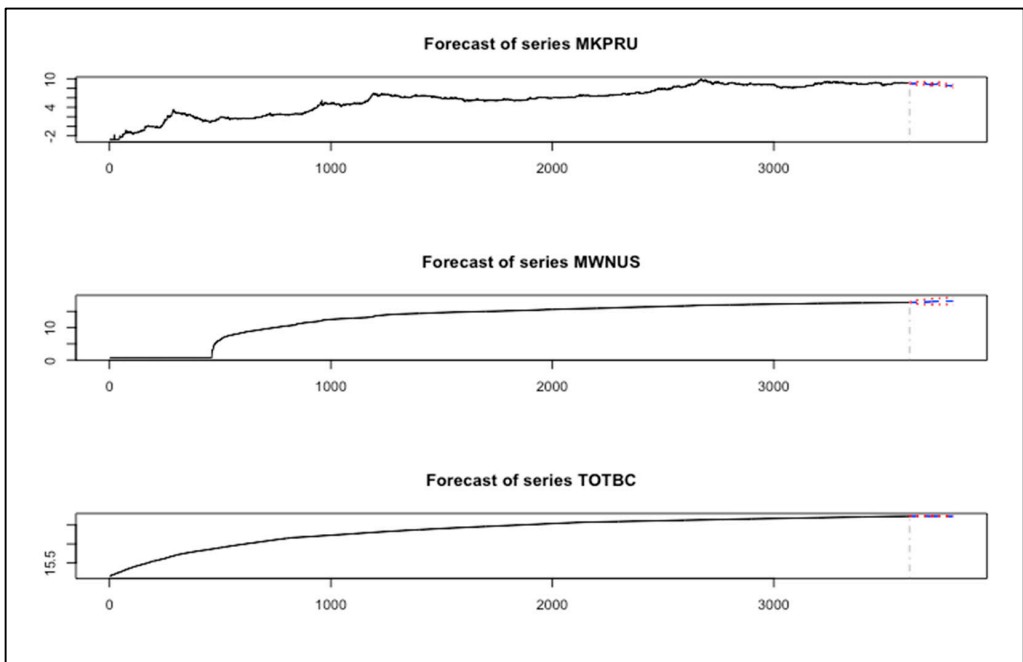

**Figure 10.** Forecasting the endogenous variables using Full timeframe data (VAR).

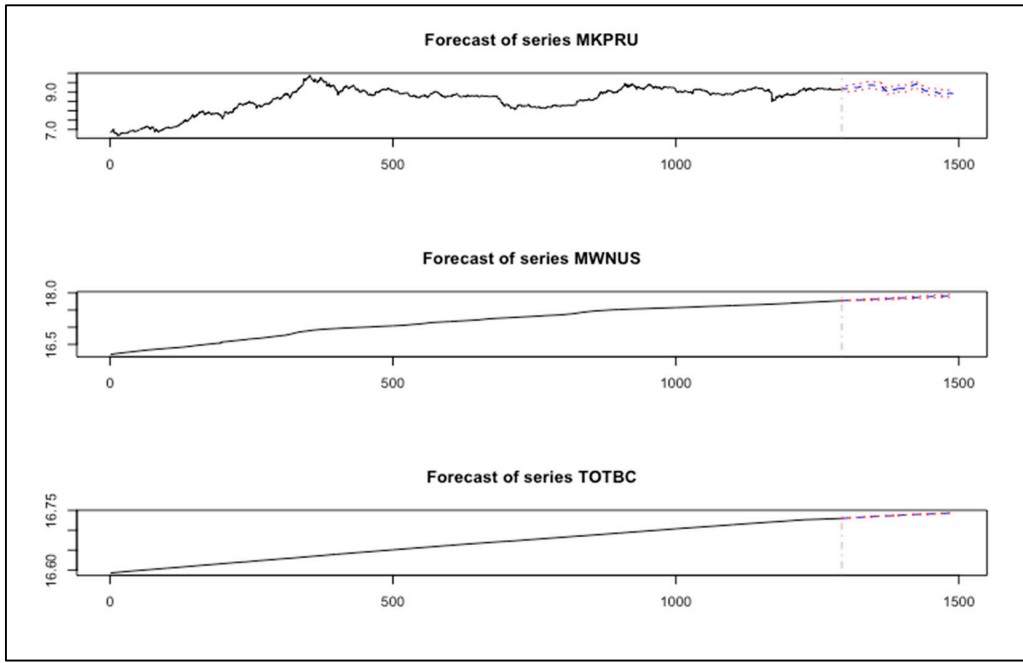

**Figure 11.** Forecasting the endogenous variables using Post-boom timeframe data (VAR).

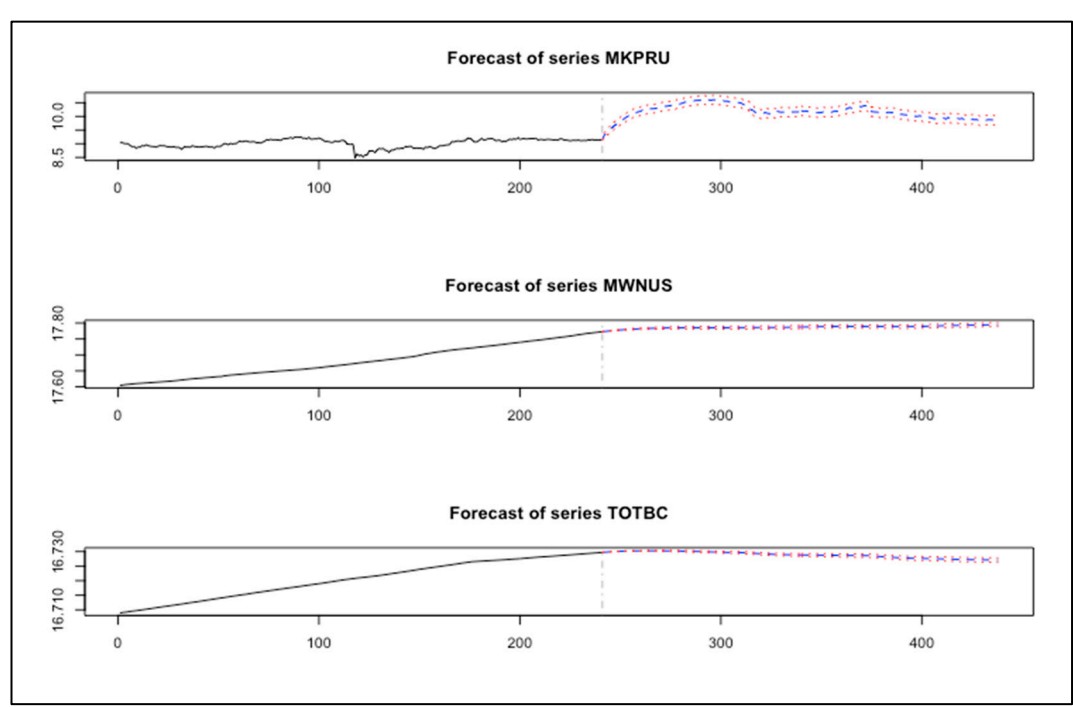

**Figure 12.** Forecasting the endogenous variables using Year of 2020 timeframe data (VAR).

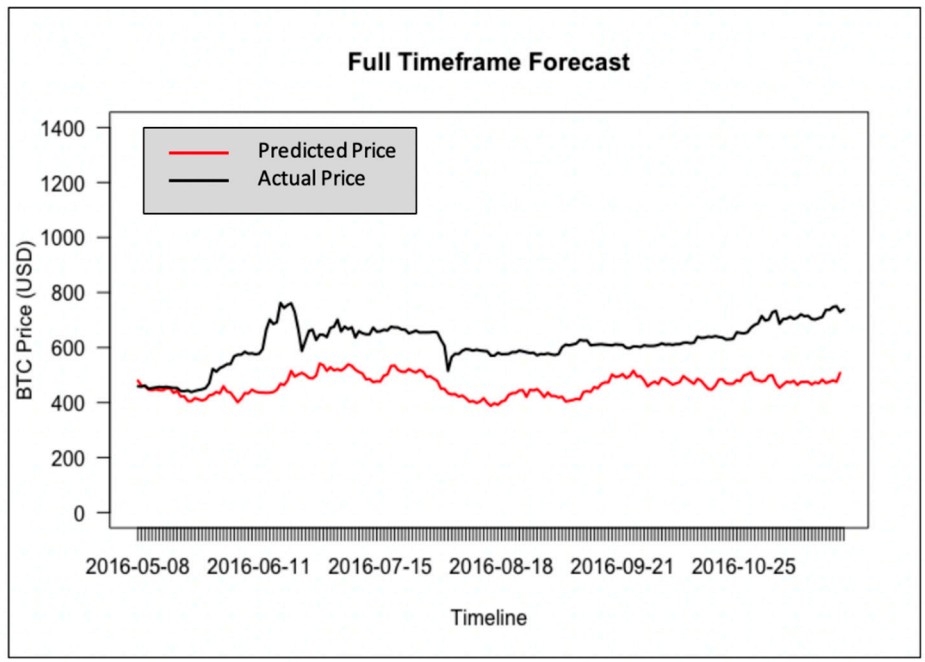

**Figure 13.** Forecasting Bitcoin closing price using Full timeframe data (BVAR).

### 5.2.3. Results of the BVAR Model: Experiment A

The forecasting results of Bitcoin price in USD for Full, Post-boom, and the Year of 2016 timeframes are shown in Figures 13–15, respectively. The red lines in each plot are from the BTC Market Price dataset (MKPRU) of Quandl. The mean absolute percentage error (MAPE) of each forecasting result was calculated to evaluate the model performance. The forecasting of Year of 2016 and Post-boom timeframes gave good performances, as the result of the Year of 2016 timeframe has a MAPE value of 2.38% and the MAPE value of the Post-boom timeframe result is 2.85%. However,

forecasting price using the Full timeframe resulted in the largest MAPE value, 19.88%. The BVAR model provided high forecasting accuracy with fewer data available or shorter timeframe in the period of [January 2009–November 2016].

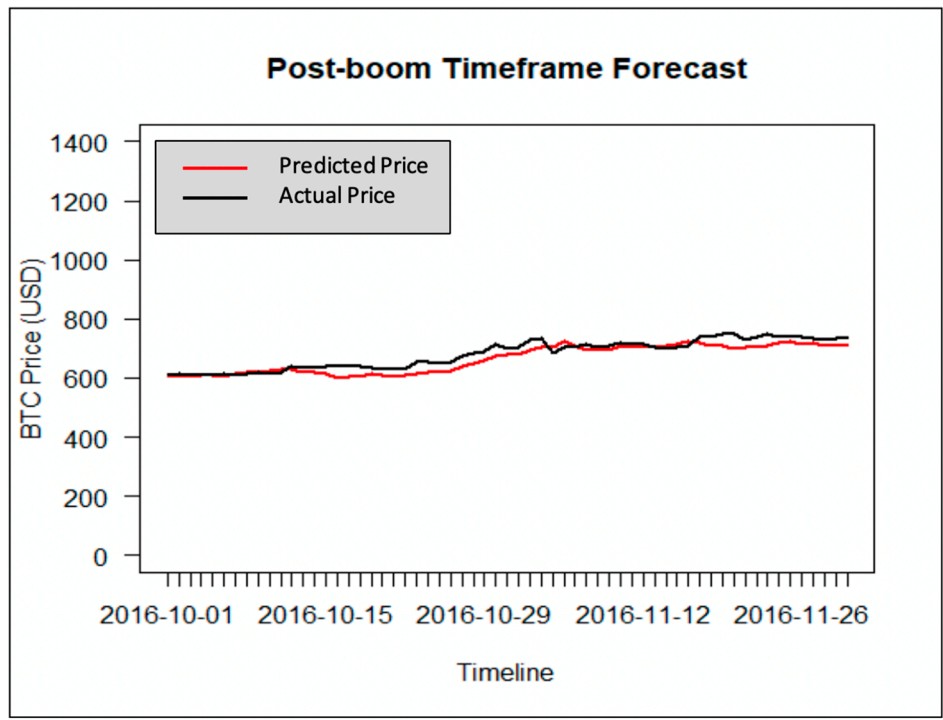

**Figure 14.** Forecasting Bitcoin closing price using Post-boom timeframe data (BVAR).

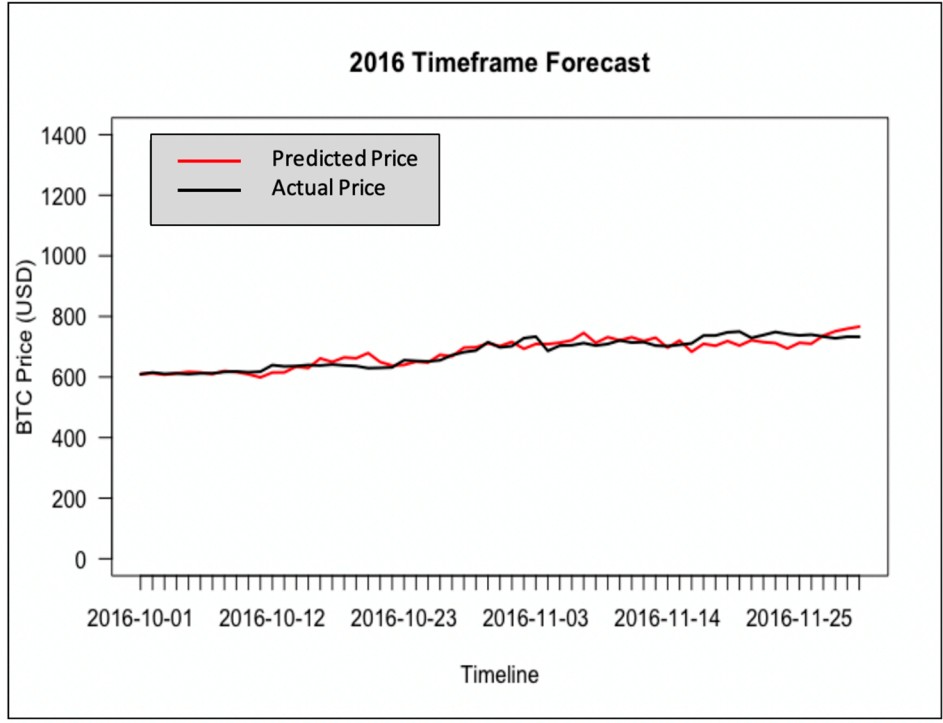

**Figure 15.** Forecasting Bitcoin closing price using Year of 2016 timeframe data (BVAR).

### 5.2.4. Results of the BVAR Model: Experiment B

In this experiment, we evaluated the performance of the VAR model using the period [January 2011–August 2020] Full timeframe data, Post-boom timeframe data [January 2017–August 2020], and the Year of 2020 timeframe data [January 2020–August 2020], as shown in Figures 16–18. We can observe that the BVAR model could predict the values of the two endogenous variables (MWNUS, and TOTBC) effectively for the Post-boom period and the Year of 2020 only, while the MKPRU variable had its best prediction for the Year of 2020 alone. This experiment confirms that the BVAR model achieves better forecasting performance for short time periods.

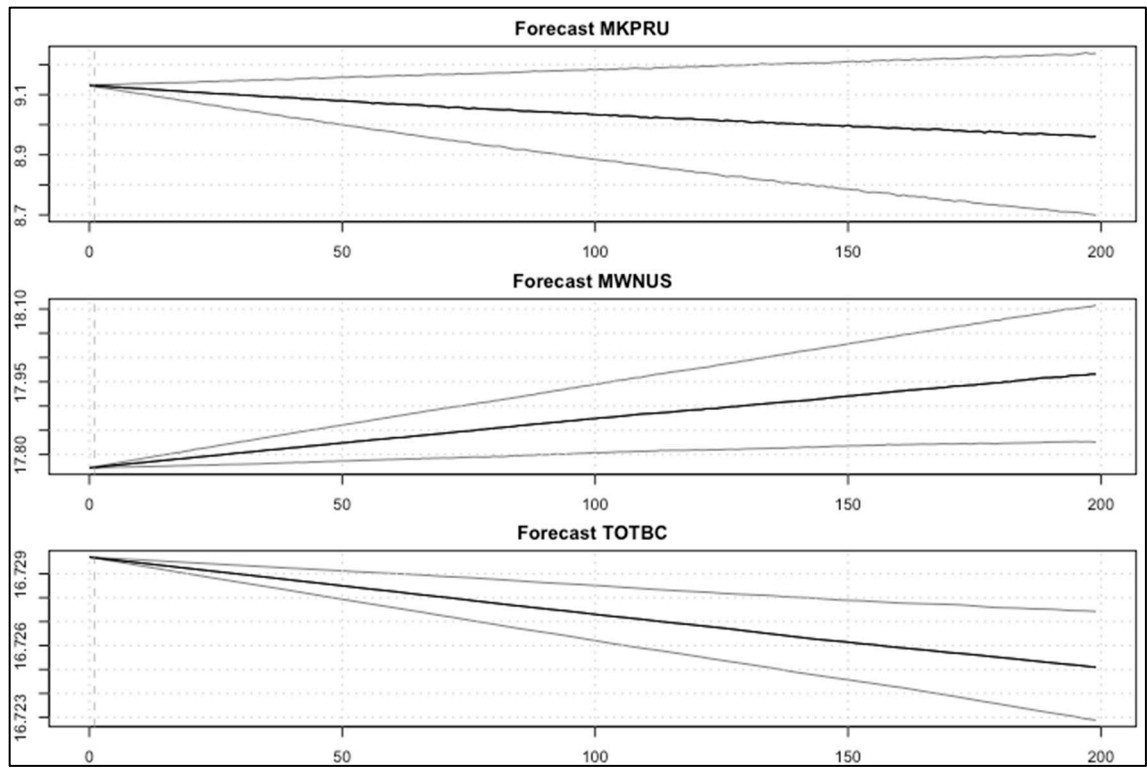

**Figure 16.** Forecasting the endogenous variables using Full timeframe data (BVAR).

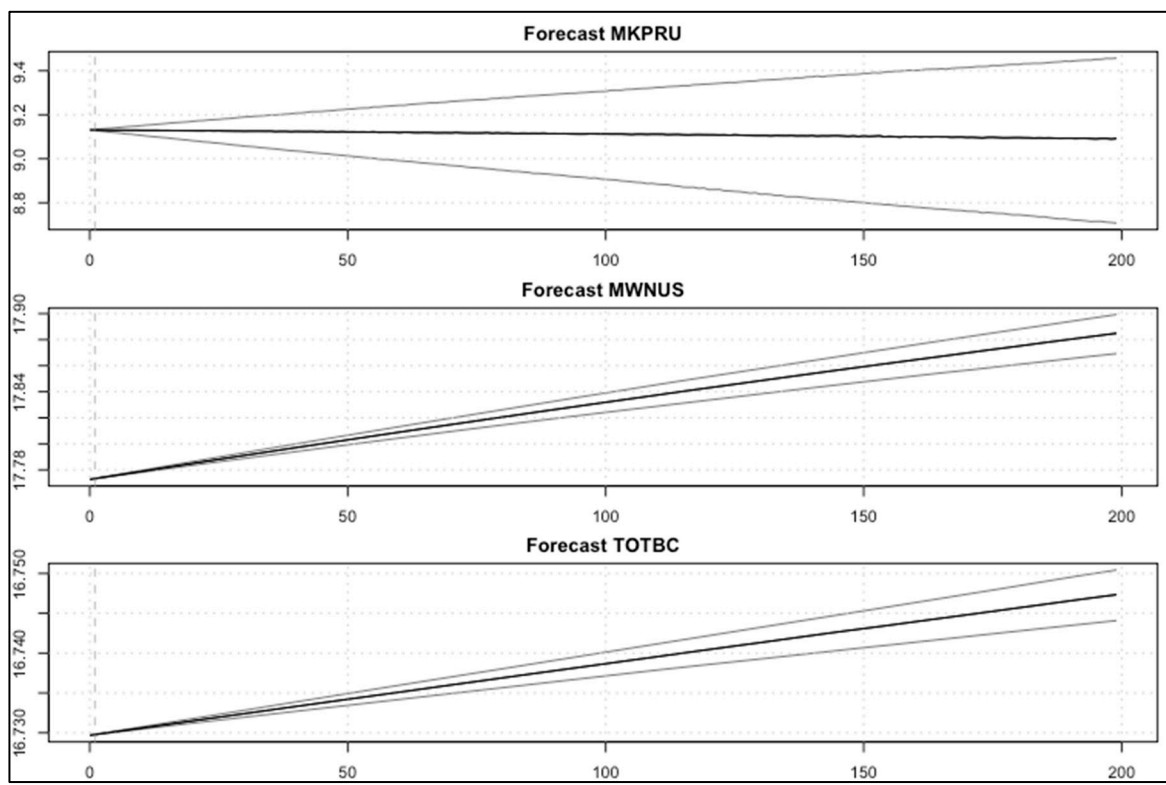

**Figure 17.** Forecasting the endogenous variables using Post-boom timeframe data (BVAR).

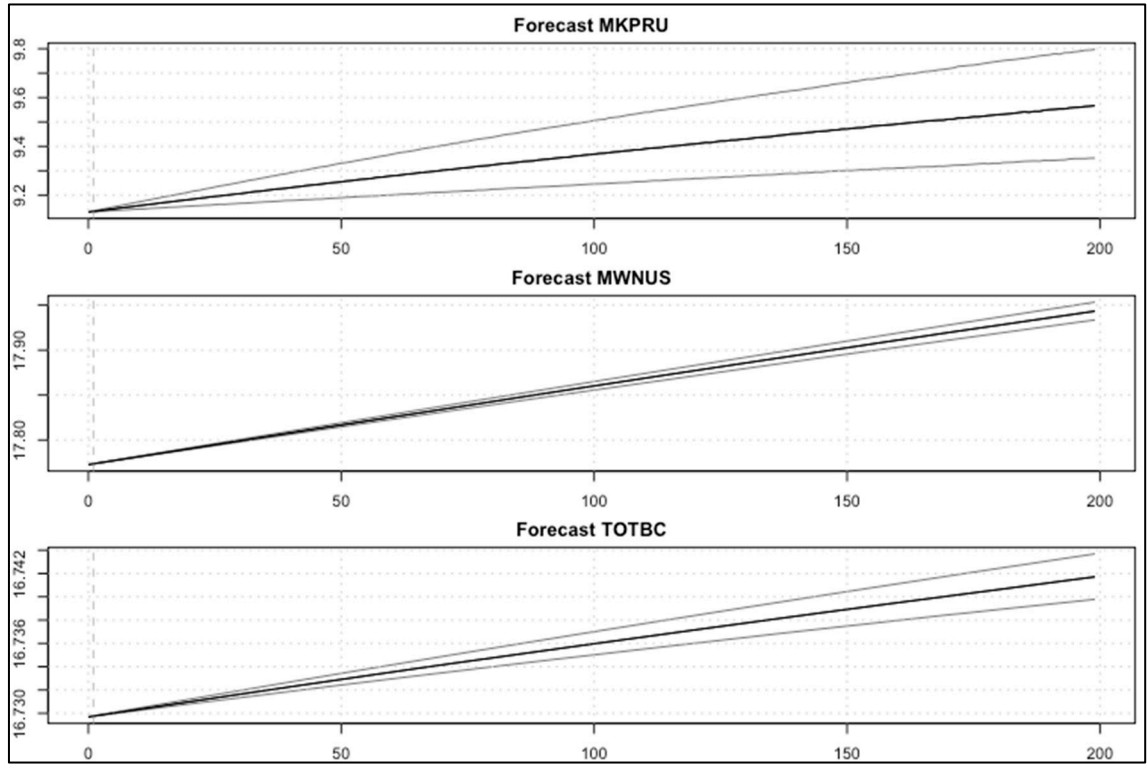

**Figure 18.** Forecasting the endogenous variables using Year of 2020 timeframe data (BVAR).

5.2.5. Analysis and Discussion of Results

For the VAR model, the price of BTC was affected by short-term lag of itself as well as the number of MyWallet users. Surprisingly, it was not affected by the supply of BTC available on the market. One explanation for this could be that the supply of BTC is limited, and as such, this value is known by speculators beforehand as a market symmetric variable. The current BTC price was positively affected by 1, 2, 4, 5, 9, 11, 17, and 20 day lags of itself. It was negatively impacted by 7, 8, 10, 12, 16, and 18 day lags of itself, as shown in Table 1.

**Table 1.** Variables of significance and their effect.

| Variables of Significance | Effect |
|---|---|
| 1, 2, 4, 5, 9, 11, 17, 20 day lag of BTC | + |
| 7, 8, 10, 12, 16, 18, day lag of BTC | − |
| 1, 4, 6, 10 day lag of MyWallet users | + |
| 2, 5, 12 day lag of MyWallet users | − |
| Miner's Revenue, BTC Difficulty, Change in the Number of unique addresses | + |
| Number of Transactions per Block, Hash Rate | − |

The effects of MyWallet users on BTC price were slightly positive overall. In terms of exogenous variables, the Miner's Revenue (+), Number of Transactions per Block (−), BTC Difficulty (+), the Change in the Number of unique addresses used (+), and Hash Rate (−) all played a significant part in estimating BTC. The $R^2$ of the model was above 99%, with F-Stats significant at a 99% confidence level, as shown in Table 2.

**Table 2.** $R^2$ and F-statistics.

| Variable | $R^2$ | F-Statistics |
|---|---|---|
| BTC Price | 99+% | 99+% |
| MyWallet User | 99+% | 99+% |
| Total BTC | 99+% | 99+% |

In addition to analyzing the individual factors of influence on Bitcoin price, the VAR model predicted a great pattern of fluctuating prices. Compared with the forecasting price curves from the VAR model, the BVAR model gave a more accurate prediction of Bitcoin price to the actual values in general. Additionally, the availability and completeness of the input data played a significant role in the performance of the VAR model, while the BVAR model achieved a great forecasting result with a low percentage error rate while using only data of the years 2016 and 2020. The results demonstrate that the BVAR model performed well for a fairly limited number of observations.

*5.3. Comparative Analysis*

In this section, we compare the performance of the VAR and BVAR models with some of the well-known autoregression and Bayesian regression algorithms, including the autoregression integrated moving average (ARIMA) (Chu et al. 2017; Hencic and Gouriéroux 2017) and Bayesian regression (BR) (Shah and Zhang 2014). ARIMA is a commonly used model to predict the price, and the model is a combination of three basic time-series models: autoregressive, moving average, and autoregressive moving average. Bayesian regression uses statistical analysis within the context of Bayesian inference rules. The comparison was made based on the values of the root mean squared Error (RMSE), the mean absolute error (MAE), and the mean absolute percentage error (MAPE) (Tan and Kashef 2019; Tobin and Kashef 2020). In this section, we focus on the data timeframe from Experiment B [January 2011–August 2020] and the variable of interest MKPRU (the equilibrium closing price of the BTC market as denominated by the US dollar). As shown in Tables 3–5, for the Full timeframe, the VAR model had the best performance. For the Post-boom timeframe, both the VAR

and the BVAR models had the lowest RMSE, MAPE, and MAE values. Finally, for the Year of 2020, the VAR and the BVAR models had better performance than the ARIMA and BR models.

**Table 3.** Accuracy of forecasting models: Full Timeframe.

|  | MAPE | RMSE | MAE |
| --- | --- | --- | --- |
| VAR | 0.0249 | 0.3102 | 0.2260 |
| ARIMA (2,2,1) | 0.0421 | 0.3900 | 0.3258 |
| BR | 0.0362 | 0.3554 | 0.3826 |
| BVAR | 0.0286 | 0.3375 | 0.2501 |

**Table 4.** Accuracy of forecasting models: Post-boom timeframe.

|  | MAPE | RMSE | MAE |
| --- | --- | --- | --- |
| VAR | 0.0248 | 0.2708 | 0.2212 |
| ARIMA (2,2,1) | 0.0421 | 0.3900 | 0.3258 |
| BR | 0.0351 | 0.3693 | 0.2776 |
| BVAR | 0.0264 | 0.2806 | 0.2286 |

**Table 5.** Accuracy of forecasting models: Year of 2020 timeframe.

|  | RMSE | MAE | MAPE |
| --- | --- | --- | --- |
| VAR | 0.0123 | 0.1235 | 0.1023 |
| ARIMA (2,2,1) | 0.0143 | 0.1908 | 0.1262 |
| BR | 0.0129 | 0.1418 | 0.1158 |
| BVAR | 0.0130 | 0.1273 | 0.1247 |

## 6. Conclusions and Future Directions

In this paper, two VAR models were developed to analyze and understand the mechanics of the BTC market. The developed models were tested in predicting the endogenous variables using selected features of exogenous variables. The two models were compared with the state-of-the-art forecasting models in order to show their efficiency. This research presents a powerful way to predict Bitcoin market price and an interesting look at what factors of this BTC network can shape new innovations in blockchain and the future of digital currency. As a new currency not administered by the government, there are many interesting behaviors that can be studied. From the perspective of miners, investors, or users of BTC, these findings may be useful for understanding the movements of the price of the BTC, and could help to understand what influence each of the exogenous factors has on the price of BTC. Future experiments for BTC prices will use a non-linear or dynamic VAR, which is suitable for BTC simulation. Dynamic VAR accounts for the change in a relationship by allowing the coefficients to change over time, which makes it much more challenging to analyze. The technical indicator could be extended as an exponential moving average or volume-weighted average price. Different priors can be suggested for future directions, such as the independent normal-Wishart. Additionally, analyzing the daily market returns in order to understand the distribution of daily behavior could provide insight into the classification of upward and downward trends. Incorporating the classification would enable research to understand price action in more depth with increasingly sophisticated machine-learning or nonlinear models. Finally, further investigation combing machine-learning prediction models is recommended.

**Author Contributions:** Software, A.I., E.V.; Supervision, R.K.; Visualization, A.I., E.V., M.L., and E.H.; Writing—original draft, A.I.; Writing—review & editing, A.I. and R.K.; Validation, A.I. and M.L.; Source—A.I., M.L., E.V., and E.H. All authors have read and agreed to the published version of the manuscript.

**Funding:** This research received no external funding.

**Conflicts of Interest:** The authors declare no conflict of interest.

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
