# Peer review of "Bitcoin Network Mechanics: Forecasting the BTC Closing Price Using Vector Auto-Regression Models Based on Endogenous and Exogenous Feature Variables"

_jrfm, doi:10.3390/jrfm13090189_

Round 1
Reviewer 1 Report
See the attached report

Author Response
Reply to Reviewer 1
Thank you for reviewing our paper entitled “Bitcoin Network Mechanics: Forecasting The Bitcoin Closing Price Using Vector Auto-Regression Models Based on Endogenous and Exogenous Feature Variable”. We are grateful for the feedback of the review team, which has led to a substantial improvement of our paper. We list the comments of the review team below (in blue italic font), followed by our responses (in black regular font).
Reviewer 1:
The paper examines the market movement and identifies the major price drivers including stakeholders through Vector Autoregressive and Bayesian VAR models. However, although the paper is well written, there are some major issues related to the empirical results. Below, I will discuss them.
Response: Thank you for your careful reading of our paper and for your valuable suggestions. We have undertaken a very serious revision according to all your suggestions. Please see below for our point-by-point response to your comments.
Reviewer 1: The estimation of cryptocurrencies is an hot topic in the financial and econometric literature in the last years. Therefore, I would like to bring the attention of the authors to some papers that are gaining importance in the econometric/finance community. From a forecasting point of view, Catania et al. (2019) and Bohte and Rossini (2019) study the forecasting performance of Cryptocurrencies by vector autoregressive model without and with time-varying volatility. On the other hand, Bianchi (2020) investigated if there is a relationship between returns on cryptocurrencies and traditional asset classes and Bianchi et al. (2020) study the relationship between the returns on stable-coins and major cryptocurrency pairs within the context of a large Bayesian Vector Auto-regressive (BVAR) model, and contribute to a growing literature that aims at understanding the role of cryptocurrency markets as alternative investments
Response: Thank you for bringing these papers into our attention, we have added them in the references section and cited them into the related work as well. We have also cited reference [10] which discusses the volatility of the BTC.
Reviewer 1: Could the authors expand the data until 2020? Looking at the data description, I was wondering why the authors do not use logarithm of the returns in order to standardized the time series. Moreover, are the data daily or weekly?
Response: We have used a daily data, and we added a new experiment, experiment B, to include the period from Jan 2011 until August 2020. In experiment A, we have data points with ZERO values, so we were not able to use logarithm scaling. However, in experiment B, our analysis includes the period (Jan 2011-August 2020) which has no zero values; thus, results are shown with the logarithm scaling. Experiment B has been added for both the VAR and the BVAR models (sections 5, 5.2, 5.2.2 and 5.2.4).
Reviewer 1: Regarding the explanatory variable Xt is it available at the same time of the observation variable yt or it is related to the previous day?
Response: in our prediction models, we define the variable , thus the Xt is collected at the same time of the observation Yt.
Reviewer 1: In Equation (5), the authors use a different specification of Yt with respect to Equation (1), why?
Response: we apologies for this writing error, this is a typo and we have corrected this in the manuscript.
Reviewer 1: Have you tried to implement a independent Normal-Wishart prior instead of the Minnesota prior?
Response: we have tried the Minnoesta prior as it is the default in our R implementation, we have used it across all models. We have also listed the use of the “independent Normal-Wishart prior” as future directions in section 6.
Reviewer 1: The Figures representing the results are really difficult to understand, could the authors provide better plots? In particular, is really difficult to understand the forecasting of the variables.
Response: we have enlarged and replaced the plots with high resolution images for better visualization, list of changes includes Fig. 4, 6, 8. We added new plots for experiment B that illustrate the forecasting of Endogenous variables for the period January 2011-August 2020. For the VAR model, the added figures include Fig. 10-13, for the BVAR model, the added plots are Fig 16-18.
Reviewer 1: Related to my previous point, the authors can provide evidence of density forecasting measures, such as the Continuous ranked probability score (CRPS) or the log predictive score. Moreover, have the authors compared the results of the frequentist versus the Bayesian models?
Response We have added various evaluation metrics such as the Root Mean Squared Error (RMSE), The Mean Absolute Error (MAE), and Mean Absolute Percentage Error (MAPE) to be consistent with the results and measures completed in experiment A. we have also added a new comparative analysis section (Section 5.3), in which we compare the performance of both the VAR and the BVAR models with the Integrated Moving Average (ARIMA) [26][27] and Bayesian Regression (BR) [31] in predicting the closing price of the Bitcoin.
Thank you once again for your careful reading of our paper and for your helpful suggestions and your encouragement. With the changes that we have made in this revision, we hope you agree that this paper has been significantly improved.

Reviewer 2 Report
The authors forecast bitcoin closing price using vector auto-regression models based on endogenous and exogenous feature variables. The findings in the paper will be useful for understanding the movements of the price of the bitcoin, as well as to what influence each of the exogenous factors has on the price of bitcoin. However I have some concerns for the paper:
- The introduction mentions about VAR and BVAR without explicitly describing the techniques. I want the authors to add one line describing the two techniques.
- Adding on to the first point, the introduction section is vague and requires more literature for the readers to understand and know the motivation for the proposed work. Consider including a gist of related work section to motivate the intent of the work.
- The related work section needs to be thoroughly improved. It lists down the existing work. However, it does not include the shortcomings of each of the papers and also how proposed techniques outperform the related work.
- VAR interest equation requires a range of the variable to help understand the relationship of beta and coefficients.
- Figure 2 can be improved by changing the resolution and fontsize.
- The results which are shown in the paper are from year 2016. Including more recent work might increase the interest of the readers (consider including Covid impact).
- Comparative analysis with existing work should be included.
- There are typographical and syntactical errors in the paper. Please proofread the manuscript thoroughly.
Author Response
Reply to Reviewer 2
Thank you for reviewing our paper entitled “Bitcoin Network Mechanics: Forecasting The Bitcoin Closing Price Using Vector Auto-Regression Models Based on Endogenous and Exogenous Feature Variable”. We are grateful for the feedback of the review team, which has led to a substantial improvement of our paper. We list the comments of the review team below (in blue italic font), followed by our responses (in black regular font).
Reviewer 2: The introduction mentions about VAR and BVAR without explicitly describing the techniques. I want the authors to add one line describing the two techniques.
Response: we have provided a brief description to both the VAR and the BVAR models in the introduction section.
Reviewer 2: Adding on to the first point, the introduction section is vague and requires more literature for the readers to understand and know the motivation for the proposed work. Consider including a gist of related work section to motivate the intent of the work.
Response: Thanks you for the comment, we have enriched and extended both the introduction (sec 1 ) and related work section (Sec 3) to introduce the motivation of the proposed work.
Reviewer 2: The related work section needs to be thoroughly improved. It lists down the existing work. However, it does not include the shortcomings of each of the papers and also how proposed techniques outperform the related work.
Response: we have extended the related work section with recent related papers. We have also outlined the shortcoming of these papers and how our paper addresses those shortcomings as shown in Sec 3.
Reviewer 2: VAR interest equation requires a range of the variable to help understand the relationship of beta and coefficients.
Response: The VAR equation is listed in Eq.5 with the list of Endogenous and Exogenous Feature Variables defined in Eq.1 and Eq.4, respectively which is then defined in the text of section 2.
Reviewer 2: Figure 2 can be improved by changing the resolution and fontsize.
Response: Thanks for the comment, we have changed the resolution and font size of each image in the paper, and we have replaced low resolution image with high resolutions images such as Fig.4, 6,8. Fig.1 is now changed with a better resolutions plot, Fig.2 is added to reflect the new experiment on the new datasets (Jan 2011- August 2020). We added new plots for experiment B that illustrate the forecasting of Endogenous variables for the period January 2011-August 2020 (for the VAR model, the added figures include Fig. 10-13), for the BVAR model, the added plots are Fig 16-18)
Reviewer 2: The results which are shown in the paper are from year 2016. Including more recent work might increase the interest of the readers (consider including Covid impact).
Response: we have added a new experiment, experiment B, to include the period from Jan 2011 until August 2020. Experiment B has been added for both the VAR and the BVAR models (sections 5, 5.2, 5.2.2 and 5.2.4).
Reviewer 2: Comparative analysis with existing work should be included.
Response: We have added various evaluation metrics such as the Root Mean Squared Error (RMSE), The Mean Absolute Error (MAE), and Mean Absolute Percentage Error (MAPE) to be consistent with the results and measures completed in experiment A. we have also added a new comparative analysis section (Section 5.3), in which we compare the performance of both the VAR and the BVAR models with the Integrated Moving Average (ARIMA) [26][27] and Bayesian Regression (BR) [31] in predicting the closing price of the Bitcoin.
Reviewer 2: There are typographical and syntactical errors in the paper. Please proofread the manuscript thoroughly.
Response: The paper has been fully proofread by a professional proof-reader and all typographical and syntactical errors are corrected.
Thank you once again for your careful reading of our paper and for your helpful suggestions and your encouragement. With the changes that we have made in this revision, we hope you agree that this paper has been significantly improved.

Round 2
Reviewer 1 Report
The authors are to be congratulated for the revision that has improved the paper. In the new version of the paper, the authors have addressed mostly all the points raised in the previous version. In particular, I am very positive about the paper and the related answer to the comments, but I have one concerned regarding the forecasting exercise.
The authors should state in the paper if they are using a direct or a recursive forecasting methods since they are forecasting y_t with respect to x_t.
Author Response
Reviewer 1:
The authors are to be congratulated for the revision that has improved the paper. In the new version of the paper, the authors have addressed mostly all the points raised in the previous version. In particular, I am very positive about the paper and the related answer to the comments, but I have one concerned regarding the forecasting exercise.
Author Response: Many thanks for your valuable comments and feedback
Reviewer 1:
The authors should state in the paper if they are using a direct or a recursive forecasting methods since they are forecasting y_t with respect to x_t.
Author Response: we have stated in the paper that we are using direct forecasting methods in the introduction section
Reviewer 2 Report
The authors have updated the paper with some of my feedback. However there are some changes which has not been included:
- The related work section still needs significant improvement. It should explicitly mention the loophole of each existing work. The section is not thorough and is incomplete. I would encourage the authors to rewrite the section again.
- Figure 3 has not been modified as per the suggestion I advised in my previous review.
- The paper figures have font size varied in each figure. Please be consistent in the figures.
Author Response
Reviewer2: The authors have updated the paper with some of my feedback. However there are some changes which has not been included
Author response: Many thanks for your valuable comments and feedback
Reviewer2: The related work section still needs significant improvement. It should explicitly mention the loophole of each existing work. The section is not thorough and is incomplete. I would encourage the authors to rewrite the section again.
Author response: We have rewritten the related work section, and references 10-20 are now added. We have classified the literature review into two main categories machine learning models and time-series forecasting models. we have discussed the shortcoming of each technique, and we have highlighted how our proposed models overcome these shortcomings
Reviewer2: Figure 3 has not been modified as per the suggestion I advised in my previous review.
Author response: All figures have been replaced with high-resolution images in the previous round, and we have even increased the size of each image for better visualization as recommended. Hopefully, it is satisfiable
Reviewer2: The paper figures have font size varied in each figure. Please be consistent in the figures.
Author response:The caption of each figure is now revised, such that all captions have the same font size.
Round 3
Reviewer 2 Report
The authors have updated the paper according to the feedback.
Just one change, for Figure 3, please use smaller font size inside the figure.